# A deep-learning system bridging molecule structure and biomedical text with comprehension comparable to human professionals

Zheni Zeng [1,2], Yuan Yao[1,2], Zhiyuan Liu [1✉] & Maosong Sun [1✉]

To accelerate biomedical research process, deep-learning systems are developed to automatically acquire knowledge about molecule entities by reading large-scale biomedical data. Inspired by humans that learn deep molecule knowledge from versatile reading on both molecule structure and biomedical text information, we propose a knowledgeable machine reading system that bridges both types of information in a unified deep-learning framework for comprehensive biomedical research assistance. We solve the problem that existing machine reading models can only process different types of data separately, and thus achieve a comprehensive and thorough understanding of molecule entities. By grasping meta-knowledge in an unsupervised fashion within and across different information sources, our system can facilitate various real-world biomedical applications, including molecular property prediction, biomedical relation extraction and so on. Experimental results show that our system even surpasses human professionals in the capability of molecular property comprehension, and also reveal its promising potential in facilitating automatic drug discovery and documentation in the future.

[1] Department of Computer Science and Technology, Tsinghua University, Beijing, China. [2] These authors contributed equally: Zheni Zeng, Yuan Yao. ✉email: liuzy@tsinghua.edu.cn; sms@tsinghua.edu.cn

Understanding molecule entities (i.e., their properties and interactions) is fundamental to most biomedical research areas. For instance, experts study the structural properties of protein molecules to understand their mechanisms of action[1], and investigate the interactions between drugs and target molecules to prevent adverse reactions[2]. To this end, people have built many biomedical knowledge bases (KBs), including PubChem[3], Gene Ontology[4], and DrugBank[5]. However, existing KBs are still far from complete due to the rapid growth of biomedical knowledge and the high cost of expert annotation. With the rapid progress of deep learning, machine reading systems are developed to automatically acquire biomedical knowledge by reading large-scale data, accelerating recent biomedical research in many cases[6].

However, compared to human learners, machine reading systems still have a huge gap in terms of both versatile reading and knowledgeable learning[7]. In the acquisition of biomedical molecule knowledge, humans are capable of **versatilely reading** different types of information that complementarily characterize molecule entities, including molecule structures and biomedical text. Specifically, molecule structures provide concise standardized internal information, where functional groups and their positions are strong indicators of molecular properties and interactions.

In comparison, biomedical text provides abundant flexible external information of molecule entities reported from wet-lab experiments. Utilizing complementary information is typically crucial for human learners to achieve comprehensive molecule understanding. Moreover, humans are able to **knowledgeably learn** and leverage meta-knowledge within and across different information—establishing fine-grained mappings between semantic units from different information sources, e.g., functional groups and natural language phrases—for deep molecule understanding.

To the best of our knowledge, all existing machine reading systems for biomedical knowledge acquisition are confined to either internal molecule structure information or external biomedical text information in isolation, and different models have to be developed to process each type of information. This limits not only the generality of machine reading systems, but also the performance of knowledge acquisition due to the intrinsic nature of each information. Specifically, information from molecule structure is concise but typically limited compared to information from wet-lab experiments, while information from biomedical text enjoys better abundance and flexibility but usually suffers from noisy extraction processes. Moreover, confined to single information sources, machine reading systems can hardly learn meta-knowledge beyond single information for deep molecule understanding. Inspired by human learners, it is desirable to build a knowledgeable machine reading system that versatilely learns from both information sources to better master molecule knowledge so as to assist biomedical research. However, it is nontrivial to jointly model the heterogeneous data in a unified framework, and challenging to learn the meta-knowledge without explicit human annotation.

In this work, we pioneer a knowledgeable machine reading system, establishing connections between internal information from molecule structures and external information from biomedical text, as shown in Fig. 1. We jointly model the heterogeneous data in a unified language modeling framework, and learn the meta-knowledge by self-supervised language model pre-training techniques on large-scale biomedical data without using any human annotation. Specifically, for molecule encoding, there are various plausible choices such as descriptor-based models[8] and simplified molecular-input line-entry system (SMILES)-based models[9]. In this work, we serialize molecule structures using SMILES for programmatic simplicity, since they can be easily unified with textual tokens and processed by the Transformer architecture. Then the SMILES representations are segmented into frequent substring patterns using the byte pair encoding (BPE)[10] algorithm in a purely data-driven approach inspired by the tokenization and encoding method of predecessors[11]. Interestingly, we observe that the resultant substring patterns are chemically explainable (e.g., carbon chains and functional groups), and can potentially be aligned to molecule knowledge distributed in biomedical text. Therefore, we insert the segmented SMILES-based representations of molecules into their corresponding mentions in biomedical papers, and model the resultant data under a unified language modeling framework. Finally, the meta-knowledge is learned via self-supervised language model pre-training on the large-scale biomedical data. After pre-training, the meta-knowledge can be readily transferred via fine-tuning to facilitate various real-world biomedical applications.

Comprehensive experiments demonstrate that, by learning deep meta-knowledge of molecule entities, the proposed model achieves promising performance on various biomedical applications in both molecule structure and biomedical text, including molecular property prediction, chemical reaction classification, named entity recognition and relation extraction. More importantly, by grasping meta-knowledge between molecule structures and biomedical text, our model enables promising cross-information capabilities. Our model is able to produce natural language documentation for molecule structures, and retrieve molecule structures for natural language queries. Such intelligent capabilities can provide convenient assistants and accelerate biomedical research. Through the multiple-choice questions about molecular properties, our model, which gets an accuracy score over 0.83, is proved to have deeper comprehension towards molecule structure and biomedical text than human professionals that get 0.77 accuracy. In the case study of six functional natural language queries towards 3,000 candidate molecule entities, 30 out of 60 retrieved entities can be supported by wet-lab experiments, among which 9 entities are not reported in PubChem (thus newly discovered), showing the promising potential of our model in assisting biomedical research in the future.

Our contributions are summarized as follows:

(1) We present a knowledgeable and versatile machine reading system that bridges molecule structures and biomedical text.
(2) Our major contribution lies in the application of the proposed model in assisting drug discovery and documentation for biomedical research.
(3) Comprehensive experiments show the effectiveness of the proposed model.

## Results

**Overview of KV-PLM**. We propose KV-PLM, a unified pre-trained language model processing both molecule structures and biomedical text for knowledgeable and versatile machine reading. KV-PLM takes the popular pre-trained language model BERT[12] as the backbone. To process the heterogeneous data in a unified model, molecule structures are first serialized into SMILES[9] strings, and then segmented using BPE[10] algorithm. To learn the meta-knowledge between different semantic units, we pre-train KV-PLM using the masked language modeling task[12]. During pre-training, part of the tokens (including tokens from molecule structure and biomedical text) are randomly masked, and the model is asked to reconstruct the masked tokens according to the context. In this way, the model can grasp the correlation between molecule structure and biomedical text without any annotated data. After pre-training, the model can be readily fine-tuned to facilitate various mono-information and cross-information biomedical applications.

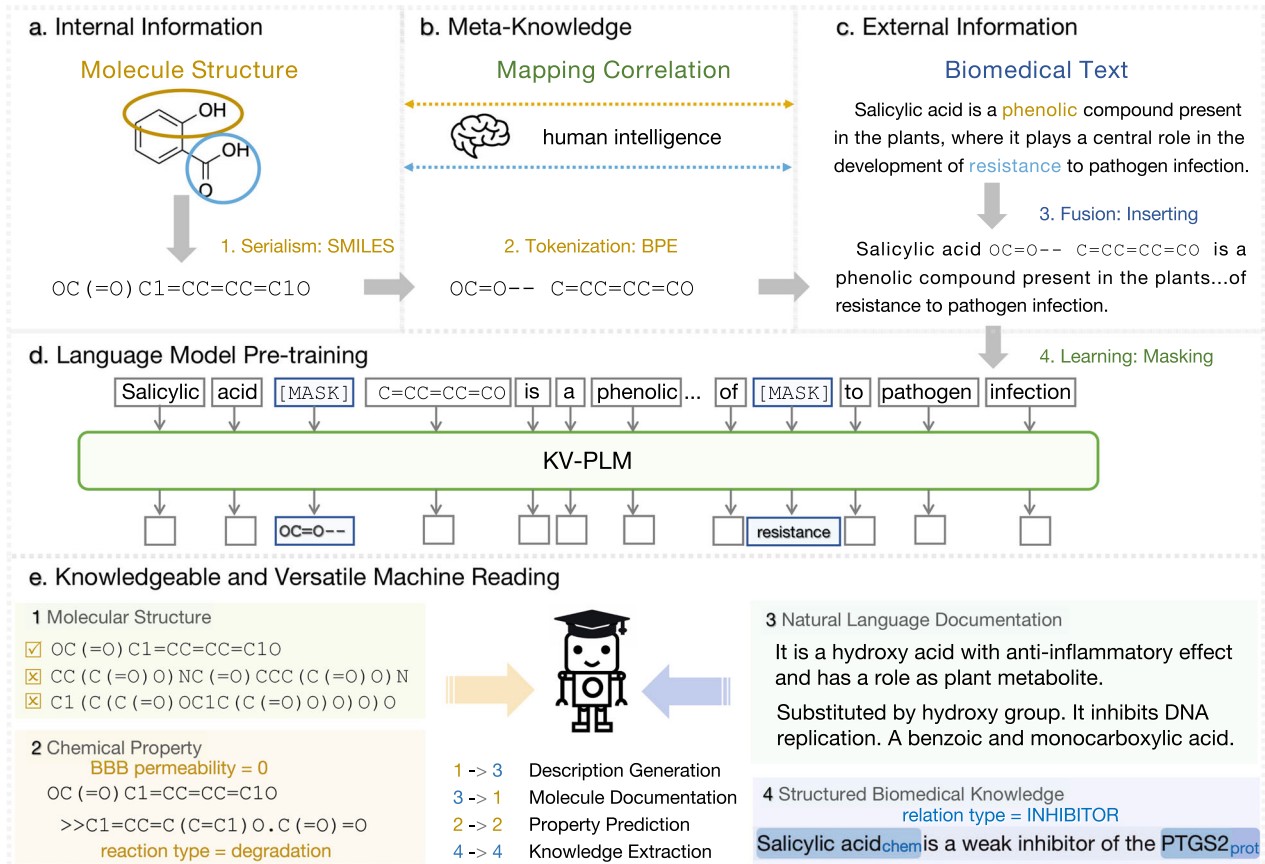

**Fig. 1 Conceptual diagram of knowledgeable and versatile machine reading.** Here we take salicylic acid as an example. Inspired by humans that versatilely learn meta-knowledge within and across different information, our machine reading system first serializes, **a** molecule structures via BPE on SMILES strings, then inserts the substrings into **c**. large-scale corpus and learns **b** fine-grained mapping between different semantic units by **d** mask language modeling. In this way, the system can perform **e** knowledgeable and versatile reading, achieving good performance on both mono-information downstream tasks and versatile reading tasks.

To comprehensively investigate the biomedical capabilities of KV-PLM, we conduct experiments in different aspects. We first evaluate KV-PLM on mono-source biomedical tasks, including molecule structure tasks and biomedical text tasks. Then we test KV-PLM on challenging versatile reading tasks that require a deep understanding of both molecule structures and biomedical text. In the following sections, we present experimental results in Table 1 from each aspect, and then draw the main conclusions. Finally, we present a case study, showing the potential of our knowledgeable machine reading system in assisting biomedical research in real-world scenarios.

**Baseline models**. We compare KV-PLM with strong baseline models to demonstrate the effectiveness of our method.

**RXNFP**. RXNFP[13] is the state-of-the-art model for chemical reaction classification. The model is based on Transformer architecture[14] and pre-trained by masked language modeling task on chemical reaction formulas. However, tailored for processing molecule structure tasks, RXNFP cannot be applied to natural language tasks.

**BERT_wo**. To observe the effect of pre-training, we adopt BERT without any pre-training as a baseline. Notice that this model tokenizes SMILES strings altogether with natural language text using the tokenizer from the frequently-used Sci-BERT model[15], thus gets piecemeal subwords which can hardly be read by humans.

**SMI-BERT**. For molecule structure tasks, one commonly-used method is to conduct mask language modeling on SMILES strings. We take SMI-BERT, which is only pre-trained on SMILES strings as a mono-information pre-trained baseline. The tokenizer is also the same as Sci-BERT.

**Sci-BERT**. One of the most frequently used pre-trained language models in biomedical domain. It is trained on plenty of natural language data and could solve natural language tasks well. In other words, Sci-BERT is also a mono-information pre-trained baseline.

**KV-PLM**. According to our idea, the model can be pre-trained on a special corpus in which SMILES strings are inserted, and in this way KV-PLM can learn mono-information knowledge. It is expected to have obviously better performance on versatile reading tasks.

**KV-PLM\***. As we have mentioned above, SMILES strings can be tokenized with a separate tokenizer and form chemically explainable substring patterns, which have no overlap with natural language tokens. We improve KV-PLM by adopting double tokenizers to process SMILES strings in a more appropriate way.

**Molecule structure tasks**. For molecule structure, SMILES strings are commonly used molecule and chemical reaction representations. We choose molecular property learning benchmark MoleculeNet[16] and chemical reaction dataset USPTO 1k TPL[13] as our experimental materials.

For **SMILES property classification task on MoleculeNet**, We choose four commonly-used classification task themes including BBBP, SIDER, TOX21, and HIV to evaluate the capability of

**Table 1 The main experimental results on mono-information tasks and versatile reading tasks.**

| Model | Molecule structure tasks | | Natural language tasks | | Versatile reading tasks | | | | |
|---|---|---|---|---|---|---|---|---|---|
| | MoleculeNet | USP-few | ChemProt | BC5CDR | S-T Acc | Rec@20 | T-S Acc | Rec@20 | Score |
| RXNFP | 65.37 ± 0.63 | 78.97 ± 3.93 | 36.60 ± 0.76 | 9.55 ± 3.05 | 1.58 ± 0.33 | 1.19 ± 0.28 | 2.26 ± 0.14 | 0.81 ± 0.05 | 25.48 ± 1.06 |
| BERT$_{wo}$ | 66.67 ± 0.29 | 33.05 ± 0.60 | 44.10 ± 5.26 | 65.69 ± 0.20 | 17.00 ± 3.06 | 0.91 ± 0.22 | 17.89 ± 2.04 | 0.74 ± 0.11 | 32.23 ± 11.6 |
| SMI-BERT | 68.61 ± 0.63 | 56.79 ± 1.40 | 46.49 ± 2.21 | 74.36 ± 0.41 | 24.68 ± 0.22 | 19.14 ± 1.05 | 22.47 ± 0.80 | 19.82 ± 0.54 | 70.08 ± 1.40 |
| Sci-BERT | 70.65 ± 0.58 | 84.50 ± 0.71 | 84.61 ± 0.58 | 89.26 ± 0.22 | 50.38 ± 1.39 | 62.11 ± 1.49 | 50.12 ± 1.67 | 68.02 ± 1.87 | 81.59 ± 0.51 |
| KV-PLM | 70.71 ± 0.32 | 85.59 ± 0.77 | 84.59 ± 0.59 | 89.00 ± 0.33 | 53.79 ± 1.42 | 66.63 ± 1.51 | 54.22 ± 0.94 | 71.80 ± 1.56 | 83.13 ± 0.31 |
| KV-PLM* | 68.34 ± 0.52 | 69.13 ± 0.46 | 85.19 ± 0.55 | 89.17 ± 0.22 | 55.92 ± 0.79 | 68.59 ± 1.03 | 55.61 ± 0.18 | 74.77 ± 0.57 | 82.39 ± 0.69 |

For versatile reading tasks, we present test accuracy and recall for both SMILES-Text retrieval and Text-SMILES retrieval on PCdes. Score stands for accuracy on the CHEMIchoice task. Boldfaced numbers indicate significant advantage over the second-best results in one-sided t-test with $p$-value <0.05, and underlined numbers denote no significant difference.

reading SMILES strings and analyzing properties of molecules. The properties these tasks focus on are blood-brain barrier penetration, the ability to inhibit HIV replication, Toxicology in the 21st Century and Side Effect Resource in order. We follow the setting recommended by the benchmark and present ROC-AUC score for evaluation. Table 1 only presents average score for the four themes. Please refer to Table 3 for more details and baselines.

For **chemical reaction classification task on USPTO 1k TPL**, it is a newly-released dataset that contains the 1000 most common reaction template classes. Previous study[13] has proved that BERT pre-trained on the large scale of SMILES strings can solve the task quite well. To make it more challenging, we generalize a few-shot learning subset (hereinafter referred to as USP-few) containing 32 training items for each class. We follow the original setting and present macro F1 for classification evaluation.

From the results we can see that pre-training greatly improves the performance. Mono-information pre-training model SMI-BERT already gets a high average score on property classification themes, showing that focusing on internal knowledge mining may finish MoleculeNet tasks quite well. Pre-training on natural language shows a further positive effect for molecule structure tasks, indicating the value of external knowledge. Sci-BERT surprisingly achieves good performance without pre-training on SMILES strings, and this leads to the assumption that there is a certain connection between SMILES patterns and natural language patterns, which is a quite interesting discovery worthy of further investigation.

Comparing KV-PLM with KV-PLM* we could see that the separate tokenizer works worse than the original natural language tokenizer on molecule structure tasks. This is because that a few atoms or functional groups and their spatial structures are ignored by the separate tokenizer for the convenience of forming substring patterns, while attention for specific atoms or functional groups is important especially for chemical reaction classification.

**Natural language tasks**. To recognize entities and extract their relations from unstructured text is a fundamental application for machine reading, and by this way we can form easy-to-use structured knowledge automatically. During the process, there are two important tasks including named entity recognition (NER) and relation extraction (RE). We choose BC5CDR NER dataset (hereinafter referred to as BC5CDR) and ChemProt dataset as our experimental materials.

For **biomedical NER task on BC5CDR**, models are required to perform sequence labeling, where each textual token is classified into semantic labels that indicate locations and types of named entities. This is an important evaluation task because entities are the main processing objects in biomedical domain, and linking between structural knowledge and raw text is also based on entity recognition. Notice that the type of entities is usually specified for biomedical NER, and BC5CDR mainly focuses on recognition for chemical molecules and diseases.

For **RE task on ChemProt**, models are required to perform relation classification for entity pairs. We expect machine reading systems to recognize the relationships between the given entities so that the raw text could be formalized into easy-to-use formats including graph and triplet. There are 13 relation classes between chemical and protein pairs. Entities are annotated in the sentences.

Results for NER and RE are shown in Table 1. We take span-level macro $F1$ score for NER and sentence-level micro $F1$ score for RE as usual. As we can see, pre-training is of key importance for natural language tasks, and cross-information pre-training achieves better performance than mono-information pre-

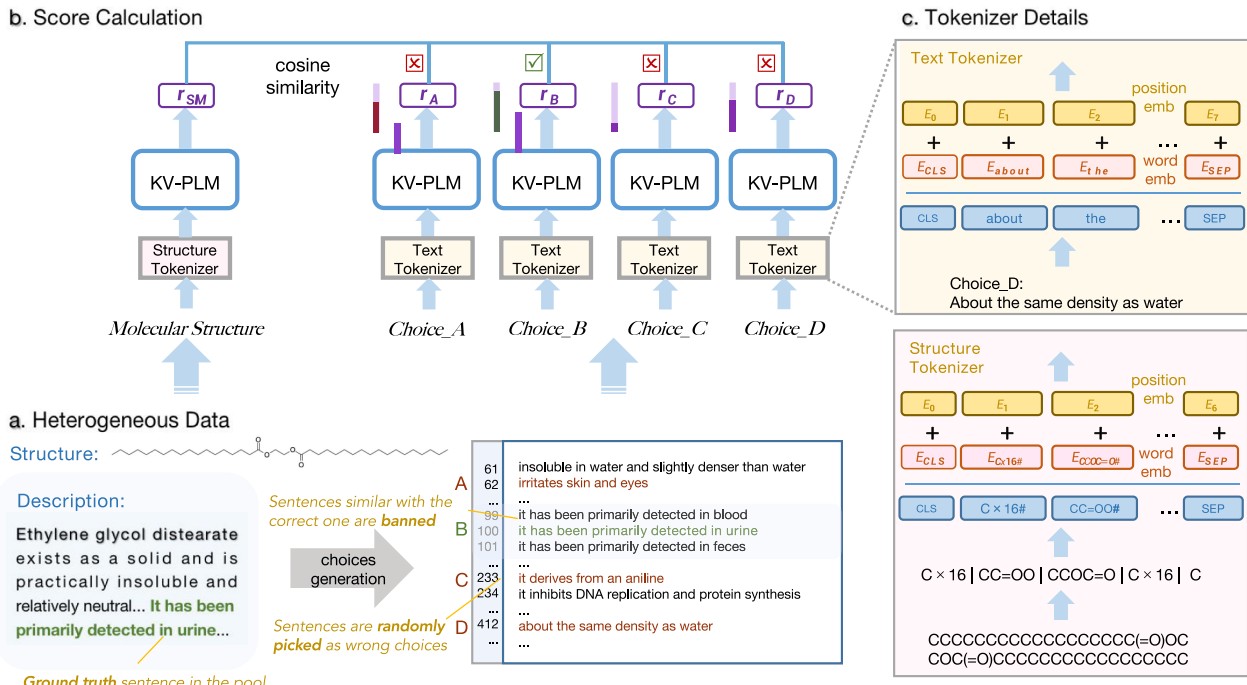

**Fig. 2 Schematic diagram for KV-PLM* finishing CHEMIchoice task.** For the given unfamiliar molecule entity, we get **a** versatile materials including structure and description, from which we know the correct sentence and randomly pick wrong sentences from the pool to form four choices. **b** Molecule structure and text of choices are fed into KV-PLM* and get their representations, based on which the confidence scores of choices are calculated by cosine similarity. **c** The tokenizers for structures and biomedical text are different. In this instance, KV-PLM* successfully finds out the correct description sentence for the given substance.

training, which proves that KV-PLM successfully learns internal structural knowledge and this can help it understand natural language. Pre-training on pure SMILE strings also helps natural language tasks, verifying the assumption that a connection exists between SMILES patterns and natural language patterns.

**Versatile reading tasks.** Since the biomedical text in natural language form is the most comprehensive material for humans, and molecule structure is the most direct information of molecules, we expect our model to process both of the two information, and understand the global and local properties of molecules.

There are few ready-made suitable datasets for versatile reading of SMILES strings and natural language documentation. We collect 15k substances in PubChem which have names, SMILES and corresponding paragraphs of property descriptions. We name our cross-information fine-tuning data as PCdes.

For **cross-information retrieval**, it is formulated as a bidirectional retrieval task for the chemical-description pairs. We evaluate the capability of understanding paragraph-level descriptions and describing global properties of molecules. KV-PLM is fine-tuned on PCdes, trying to pick the best match SMILES string or property description sentence for each other. The matching score is obtained by the cosine similarity of text representations. For evaluation metrics, we report the accuracy of the top retrieval result in randomly sampled mini-batches (64 pairs in each mini-batch). Models are also required to rank the average matching score for all the 3k molecules and description paragraphs. We present *recall@20* for both directions.

For **match judging**, we evaluate the capability of understanding sentence-level descriptions and distinguishing the local properties of molecules. To this end, we propose the multiple-choice task CHEMIchoice.

Based on descriptions in PCdes, 1.5k multiple choices are automatically generated. For the given SMILES string of substance in the test set, there are four choices of a single description sentence. Negative samples similar to the positive sample are removed, helping decrease the possibility of false negative for ground-truth answers. The system is required to choose the correct answer just like a student completing an exam, which is a quite realistic situation. The schematic diagram for CHEMIchoice is shown in Fig. 2.

We report the results of the experiments above in Table 1. Distinct data samples are used in repeating experiments with the random generation process of CHEMIchoice. As expected, with the help of cross-information pre-training on heterogeneous data, KV-PLM* can process versatile reading tasks well and achieve the best performance on most of the metrics.

For **human professional performance**, we recruited six undergraduates and postgraduates from top universities who major in chemistry without exam failure record. Given 200 questions randomly sampled from CHEMIchoice, they are required to choose the best match property description sentence for each chemical structure.

Human professionals are told that they are participating in a study to provide human performance evaluation and the experimental remuneration is determined by the rationality of their answers, thus they would not deliberately lower their level. All participants gave informed consent for their test data for this study. This research does not involve ethical issues. Academic Committee of the Department of Computer Science and Technology of Tsinghua University approved the protocol.

The performances of the six professionals exhibit diversity as shown in Fig. 3. The average score of them is 64.5 and the highest score is 76.5. We take the highest score to represent the human level since it shows the property prediction capability of an expert who is well-trained and has abundant knowledge about this type of questions, while the score is still significantly lower than our model performance. We analyze their incorrect answers and find

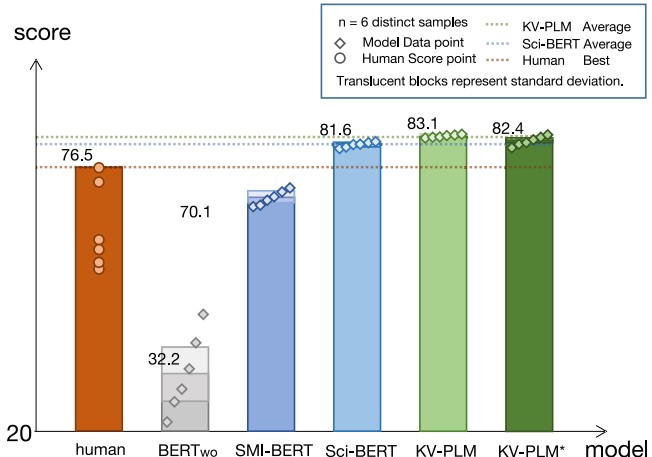

**Fig. 3 Score comparison of CHEMIchoice task.** Our model successfully surpasses human professionals, showing its promising capability of comprehending molecule structure and biomedical text. Error bars indicates standard deviation over six runs.

that human professionals tend to choose common property descriptions that do not necessarily match the target substance (e.g., irritates skin, eyes, and mucous membranes), and they are not strong in judging the unique properties of the substance to be analyzed.

**Result analysis.** From the experimental results in Table 1, we draw three main findings as below:

(1) Pre-training on mono-information data can greatly improve model performance on corresponding downstream tasks. Specifically, SMI-BERT outperforms BERT$_{wo}$ on molecule structure tasks, and Sci-BERT works better than BERT$_{wo}$ on natural language tasks. In addition, mono-information pre-trained models also achieve reasonable performance on versatile tasks. The results show that pre-training can effectively grasp meta-knowledge within each type of information to help biomedical tasks.

(2) Interestingly, we find that mono-information pre-training also brings improvements to downstream tasks from other information types. Specifically, despite being pre-trained on natural language data, when fine-tuned on molecule structure tasks, Sci-BERT even outperforms strong SMI-BERT and RXNFP models that are tailored for and pre-trained on molecule structure data. This indicates that there may exist certain connections between the patterns of molecule structures and natural language. For example, compositionality and hierarchy are important properties of both molecule structures and natural language, which can be transferred to help tasks from different information sources.

(3) Cross-information pre-training enables unified machine reading systems that outperform the baseline methods on biomedical tasks from both information sources. Moreover, our models also achieve state-of-the-art performance on versatile tasks, showing their promising potential in assisting biomedical research in these significant scenarios in the future. The results show the importance of integrating both internal and external molecule informa-tion, and the effectiveness of the proposed machine reading method for biomedical tasks.

**Case study.** In this subsection, we first give observations about the properties of substring patterns learned by models. From Fig. 4 we can see that substring patterns, which we think are of

similar properties tend to have closer fingerprints due to pre-training, showing that mask learning helps model build mapping correlation in an unsupervised fashion.

The clusters become tighter after being given the alignment supervised information just as the lower subgraph shows. More-over, we can look at the vectors in purple and find that the model can correctly distinguish between alcohol and phenol, and also understand the meaning of acid and organic salt. This proves the capability of our model to learn not only isolated but also combined properties of substring patterns and mapping between SMILES and text when finetuned on versatile reading tasks.

Further, we mainly discuss the KV-PLM* fine-tuned on PCdes due to the novelty of versatile reading tasks. To observe the retrieval capability and further potential, we can conduct both description retrieval and molecule retrieval.

For **description retrieval**, the system finds appropriate descriptive sentences and generates a paragraph of natural language description for the given SMILE string. Sentences and substances are randomly selected from PCdes test set. Figure 5 shows the property description of Tuberin predicted by KV-PLM*. The ether bond is predicted as alcohol at first, and successfully recognized as aromatic ether after getting the input benzene ring pattern. The model even predicts that Tuberin has a role as an antioxidant mainly due to the double bonds, which is not recorded in PubChem. Crystalline is also a correctly predicted property.

Another instance is 4-hydroxychalcone. Aromatic and benzoic properties are predicted after the phenol group is shown. Fruity taste and relatively neural are newly supplemented properties when given the double bond. After seeing the whole structure, the system gives out a more precise property description, predicting that it has a role as a plant metabolite and inhibitor, and also prevents oxidation.

Simpler compounds are also tested. For Chloroacetonitrile, the carbon nitrogen triple bond helps predict that it is toxic by ingestion. Combining with the chlorine the model eventually knows that it is a colorless toxic gas and has a role as an organic pollutant.

For **molecule retrieval**, the system reads natural language instructions and retrieves matching SMILES strings in turn. We require our model to find an anti-inflammatory agent from PCdes test set, and ten substances with the highest similarity scores are listed in Table 2. Most of them are related to inflammation or the immune system, and there are four substances clearly proved to be an anti-inflammatory agent. For Elocalcitol and Marinobufa-genin, data in PubChem doesn't show this information, or to say, the two agents are "newly-discovered".

Other queries including antineoplastic agent, antioxidant agent, herbicide, dye, and antidepressant drug are tested, and half of all substances the model retrieved are sure to meet the requirements. There are also several properties for substances that are missed in PubChem. For those newly-discovered molecule entities, supporting details can be found in corresponding references.

Results above show that our model catches the separation and combination properties of SMILES substring patterns, and aligns the semantic space of SMILES strings and natural language quite well. There is a chance for our method to contribute to open property prediction of molecules and drug discovery process.

## Discussion

In this article, we show the possibility of bridging the SMILES string and natural language together and propose the BERT-based model KV-PLM for knowledgeable and versatile machine reading in the biomedical domain. Through pre-training on the

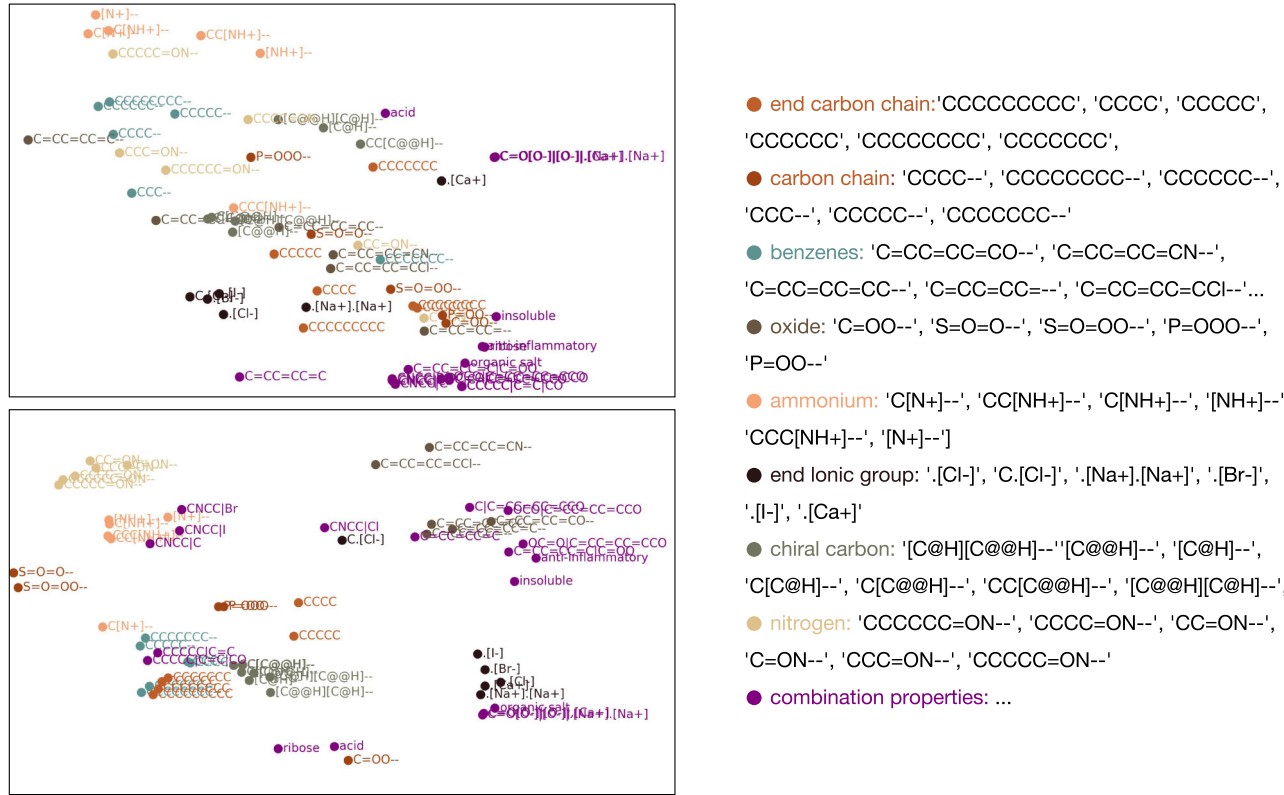

**Fig. 4 Visualizing substring pattern embeddings using t-SNE**[62]**.** Parts of substring pattern fingerprints are randomly chosen and processed for dimensionality reduction. Similar substring patterns are marked in the same colors. The upper one shows fingerprints from pre-trained KV-PLM*, and the lower one is from the model finetuned on PCdes.

special corpus, external knowledge from language and internal knowledge from molecule structure can fuse with each other unsupervisedly. KV-PLM has a basic understanding of molecule entities, and the satisfying performance when fine-tuned on various downstream tasks proves the effectiveness of molecular knowledge. Our model achieves higher accuracy than baseline models on MoleculeNet benchmark, and brings significant improvement for the more challenging task on USP-few. Even as a plain language model, our model can process classical tasks including Chemprot RE and CDR NER quite well. KV-PLM shows its capability to be a general biomedical machine reading model.

Meanwhile, the advantages for bridging the two formats of text are not restricted to the applications in mono-information form. Since there exists a correspondence between SMILES strings and natural language descriptions, we process them with a method similar to cross-information learning. By fine-tuning on PCdes data, KV-PLM can achieve cross retrieval between substances and property descriptions. We propose a new task CHEMIchoice to evaluate the reading ability on SMILES strings and natural language and also the alignment ability between them. Further, we take qualitative analysis about the potential of KV-PLM on open property prediction and drug discovery.

Still, there are some problems waiting for us to solve. First we need a better definition and evaluation for cross-domain reading tasks. Considering that models may only rely on a few sentences if training by paragraphs, we align the SMILES strings and descriptions by sentence. However, this method brings noises because randomly picked negative samples from other paragraphs may also be correct for the given substances. Besides, we simplify the SMILES strings to get more concise substring pattern results, while by removing brackets and number labels we lose information about the spatial structure. What's more, it is a simple and

rude way to linearly connect SMILES strings and natural language in series. More clever structures for fusing internal and external knowledge about chemicals and other types of entities are expected to be proposed.

Our future work will focus on the problems above, trying to get a more perfect model structure, training method and also benchmark. Graph structure and more complicated molecule representations may be adopted. Generation systems instead of retrieval may also bring different but interesting effects.

## Methods

**Related work.** Various structural machine reading systems have been developed to read molecule structures for molecular knowledge acquisition. In early years, machine learning algorithms help with molecular dynamics simulation[17,18] and energy calculations[19]. Recently, neural networks have been one of the most popular tools for analyzing molecule properties. Molecule fingerprints computed by neural networks have achieved competitive performance as compared with expert-crafted descriptors[8,20,21]. Notably, recent studies show promising results in modeling serialized molecule structures using powerful neural language models[11,22–24].

Since it is nearly impossible for human experts to read such a huge number of papers, machine reading systems, powered by natural language processing (NLP) techniques, are developed to extract molecule entities and their relations by reading large-scale biomedical literature[25–30]. To this end, researchers have proposed various neural language models to understand biomedical text, including convolutional neural networks, recurrent neural networks, recursive neural networks, and self-attention-based neural networks[31–33]. Recently, neural language models equipped with self-supervised pre-training techniques[34,35] have greatly pushed the state-of-the-art on a broad variety of biomedical information extraction tasks[15,36].

In this work, we bridge molecule structures and biomedical text in a unified multimodal deep learning framework. Previous works explore employing deep learning models to connect multimodal information, including medical images and text[37], natural images and text[38], molecules and reactions[39] and molecules and protein sequences[40]. There are also some works investigating pre-training vision-language models[41,42]. In comparison, our model jointly learns molecule structure and biomedical text representations, and establishes convenient interaction

| Molecular Structure | SMILES sub-groups | Property Prediction |
|---|---|---|
| | CO-- C=CC=CC=C-- C=C-- N-- C=O | colorless gas; toxic by ingestion; decomposed when heated; substituted by methyl group; an alcohol; inorganic |
| | CO-- C=CC=CC=C-- C=C-- N-- C=O | weakly acidic; pleasant sweet odor; found in fruits and herbs; aromatic ether; contains methoxy groups; member of benzenes |
| | CO-- C=CC=CC=C-- C=C-- N-- C=O | inhibits DNA replication; plant metabolite; overexpressed in cancer cells; aromatic ether; aromatic compound; member of benzenes |
| | CO-- C=CC=CC=C-- C=C-- N-- C=O | inhibitor; yellow crystalline; antifungal agrochemical; aromatic compound; organic compound; aromatic ether |
| COC1=CC=C(C=C1)C=CNC=O | Tuberin: It has a role of inhibitor.It is a yellow crystalline. It has a role as antioxidant. It is an aromatic compound. It is an organic compound.It is an aromatic ether. | |
| | C=CC=CC=CC=O-- C=C-- C=CC=CC=CO | toxic by ingestion; irritate skin; decomposed when heated; bacterial metabolite.; aromatic compound; benzoic acid; organic compound. |
| | C=CC=CC=CC=O-- C=C-- C=CC=CC=CO | relatively neutral; a bitter taste.; irritate skin; a fruity taste; aromatic compound; benzenes; organic compound. |
| C1=CC=C(C=C1)C(=O)C=CC2=CC =C(C=C2)O | 4-hydroxychalcone: It has a role as plant metabolite. It is a protein kinase inhibitor.It has a role as prevent oxidation and free radical formation.It is an aromatic compound, a member of benzenes. | |
| | CC-- #N-- Cl | colorless gas; toxic by ingestion; decomposed when heated; colorless liquid; substituted by methyl group; an alcohol; inorganic. |
| | CC-- #N-- Cl | colorless gas; toxic by ingestion; decomposed when heated; colorless liquid; substituted by methyl group; inorganic. |
| C(C#N)Cl | Chloroacetonitrile: It is a colorless gas.Very toxic by ingestion.It has a role as a persistent organic pollutant.Substituted by methyl group.It is a chloride compound.It belongs to chloride. | |

**Fig. 5 Case study for property prediction.** The molecular structures are first serialized in SMILES strings. With more SMILES sub-groups provided (in purple), the model can predict the properties more precisely.

| Table 2 Case study for drug discovery. | |
|---|---|
| **Property query** | **Substances retrieval result** |
| Anti-inflammatory | Effective: Elocalcitol[a 53], Fluocinolone, Fluocinonide, and Marinobufagenin[a 54] |
| | Unclear: Eribulin mesylate, U46619, Cholesteryl linoleate, Hallactone B, Leukotriene A4, and Npvvhffknivtprtppps |
| Antineoplastic | Effective: Rebeccamycin[a 55], Idarubicin, Eribulin mesylate, Piroxantrone, and XC-302 free base |
| | Unclear: Trimethoprim, Cyclomontanin C, Hexamidin, Fosinopril, and Dabigatran |
| Antioxidant | Effective: Purpurin[a 56], Aromadendrin, Amburoside A, Dioxinodehydroeckol[a 57], and hematein[a 58] |
| | Unclear: Capensinidin, 2'-Hydroxygenistein, Hydramacrophyllol A, 23566-96-3, and Olivomycin |
| Herbicide | Effective: Guanabenz[a 59], 17254-80-7, Bentazone, and Triclopyr |
| | Unclear: Diflubenzuron, Fenhexamid, 1-Azakenpaullone, Pteroic acid, and Bromhexine, C5H11ClHgN2O2 |
| Dye | Effective: Azocarmine G, Acid roseine, Acid green 3, Basic violet 14, Evans blue, Ponceau S, CHEBI:52122[a 60], Azure A, and Mercurochrome |
| | Unclear: Acid Green 50 parent |
| Anti-depressant | Effective: Benzphetamine, Benzylpiperazine[a 61], Norpramin |
| | Unclear: Dimethylaniline, 2627-86-3, Reduced Pyocyanine, 1672-76-0, 261789-00-8, Dadpm, and D extroamphetamine |

Effective substances are proved with clear documentation.
[a]Represents newly-discovered about which we list references for details.

channels between biomedical molecule knowledge and researchers for comprehensive biomedical research assistance.

**Corpus**. Our pre-training corpus comes from S2orc[43] which is a PDF-parse English-language academic papers corpus.

We take over 0.3 million papers which contain 1 billion tokens for pre-training. 75% of the papers are under *Medicine*, *Biology* or *Chemistry* fields, and the other 25% are under *Computer Science* field. In order to reduce the number of special characters related to experimental data in the text, we choose the abstract, introduction, and conclusion sections of the papers. No other special preprocessing is applied. For chemical substances we use documents from PubChem[3], in which there are over 150 million chemicals with SMILES strings and synonyms. To insert SMILES strings for chemicals, we need to do entity linking for the corpus. Since high precision and comparably low recall is acceptable for the large scale of unsupervised data, we first recognize possible entities with the help of SciSpacy[44], and then link these words with KB entities if the words can exactly match the high confidence synonyms. Notice that some substances have the same name with common objects, including "dogs", "success" and so on. Thus we filter out common words from the synonym dictionary.

There are altogether 10k chemicals with 0.5 million times of occurrence being detected in our corpus.

For the natural language part, we use the vocabulary list exactly the same as Sci-BERT, which is more appropriate than the original BERT vocabulary on academic papers. For the SMILES strings part, we apply the BPE[10] encoding method (https://github.com/rsennrich/subword-nmt) to 20,000 SMILES strings randomly chosen from PubChem and get a special vocabulary list. It is already stated that brackets and number labels are ignored. Finally, we filter out those whose frequency is lower than 100 and get 361 substring patterns among which functional groups can be observed, indicating the effectiveness of the splitting method. All the SMILES strings are split into substring patterns separately with natural language.

**Dataset**. For SMILES strings processing tasks, we adopt MoleculeNet[16], a wide standard benchmark where molecule properties for SMILES string are concluded into 17 types, and expressed in the form of classification or regression tasks. We adopt four representative tasks from MoleculeNet including BBBP, HIV, Tox21,

**Table 3 Experiment results on 4 MoleculeNet themes. Baseline results are cited from ref. [50].**

| Model | BBBP | HIV | SIDER | TOX21 | Average |
|---|---|---|---|---|---|
| D-MPNN | 71.2 ± 3.8 | 75.0 ± 2.1 | 63.2 ± 2.3 | 68.9 ± 1.3 | 69.6 |
| RF | 71.4 ± 0.0 | 78.1 ± 0.6 | 68.4 ± 0.9 | 76.9 ± 1.5 | 73.7 |
| DMP | 78.1 ± 0.5 | 81.0 ± 0.7 | 69.2 ± 0.7 | 78.8 ± 0.5 | 76.5 |
| RXNFP | 68.49 ± 0.71 | 73.46 ± 1.03 | 54.07 ± 1.64 | 65.46 ± 0.47 | 65.37 ± 0.63 |
| BERT$_{wo}$ | 68.37 ± 0.48 | 69.39 ± 1.04 | 60.19 ± 1.19 | 68.75 ± 0.64 | 66.67 ± 0.29 |
| Sci-BERT | 74.94 ± 1.30 | 75.38 ± 1.18 | 60.55 ± 2.36 | 71.72 ± 0.60 | 70.65 ± 0.58 |
| SMI-BERT | 71.12 ± 2.24 | 73.88 ± 1.45 | 59.84 ± 0.88 | 69.61 ± 0.44 | 68.61 ± 0.63 |
| KV-PLM | 74.61 ± 0.92 | 74.00 ± 1.16 | 61.51 ± 1.47 | 72.71 ± 0.59 | 70.71 ± 0.32 |
| KV-PLM* | 71.97 ± 0.85 | 71.84 ± 1.36 | 59.78 ± 1.53 | 69.79 ± 0.45 | 68.34 ± 0.52 |

and SIDER. We use the official training, validation and test sets provided by DeepChem[45] package to ensure that the performance is relatively stable and reproducible. For the two multilabel datasets Tox21 and SIDER, we report the average scores for all the tasks.

Specifically, we adopt the following tasks and datasets:

(1) *BBBP*, the blood-brain barrier penetration dataset. It includes binary labels for 2053 compounds on their permeability properties. Binary labels for penetration/non-penetration are given.

(2) *SIDER*, the Side Effect Resource database of marketed drugs and adverse drug reactions. It groups drug side effects into 27 system organ classes and includes binary labels for 1427 drugs. Symptoms waited for binary classification including endocrine disorders, eye disorders and so on.

(3) *Tox21*, a public database measuring the toxicity of compounds created by the "Toxicology in the 21st Century". It contains qualitative toxicity measurements for 8014 compounds on 12 different targets, including nuclear receptors and stress response pathways. Molecules are supposed to be classified between toxic and nontoxic on each target.

(4) *HIV*, a dataset introduced by the DTP AIDS Antiviral Screen. It tests the ability to inhibit HIV replication for 41,127 compounds required to classify between inactive and active.

Besides, we adopt USPTO 1k TPL dataset (https://github.com/rxn4chemistry/rxnfp) and create a few-shot subset. The original set has 410k data items. We randomly pick 32 items for each class and get 32k items in total. In prepossessing SMILES representations, to prevent sparse SMILES tokenization (i.e., producing over-specific infrequent tokens), we remove numbers and brackets before feeding them to KV-PLM* tokenizer. No other prepossessing steps are conducted.

For natural language processing, we adopt Chemprot and BC5CDR dataset. Chemprot is a text mining chemical-protein interactions corpus with 13 classes of relation types including *inhibitor*, *product-of* and so on. There are 1020 abstracts (230k tokens) for train set, 612 abstracts (110k tokens) for dev set and 800 abstracts (180k tokens) for test set. BC5CDR is a chemical-disease relation detection corpus with 1500 abstracts in total and equally divided into train set, dev set and test set. There are over 5k mentions of chemicals in each set. More researches focus on the NER task than the relation detection task of BC5CDR. We use the version of the two datasets provided by Sci-BERT (https://github.com/allenai/scibert) and there is no special preprocessing of the data.

For cross-information tasks, we evaluate our retrieval model on PCdes. Specifically, we substitute all the synonyms of the ground-truth substances into the word *it* to avoid information leakage. 15k SMILES-description pairs in PCdes are split into training, validation and test sets with ratio 7:1:2. For matching judging, we first construct a choice base consisting of the 870 description sentences that occur more than five times except for derivation descriptions. Then we sort the sentence strings to assign similar sentences with closer index. To generate multiple-choice questions for CHEMIchoice, for each of the 1428 test substances, we randomly choose a sentence from the corresponding ground-truth descriptions as the positive choice. The negative choices are sampled from the choice base, where the difference between indexes of positive and negative sentences is greater than 10. In this way, we largely avoid false negative choices.

**Model**. KV-PLM is based on the BERT model, which is one of the most popular language models in recent years. Specifically, KV-PLM has 12 stacked Transformer layers with 110M parameters in total, where each Transformer layer consists of a self-attention sub-layer followed by a feedforward sub-layer.

There are plenty of ready-made frameworks for BERT. For computation efficiency, we initialize our model by Sci-BERT uncased version. To adapt to downstream tasks, following previous works[12,15], we introduce a classification layer on top of the model, which can perform sequence classification and sequence labeling for SMILES string classification, RE and NER tasks.

To apply deep models to SMILES strings processing, there are different strategies for tokenization[46]. In one of our model variants KV-PLM, we directly take the tokenizer of Sci-BERT to tokenize SMILES strings, regarding them exactly

**Table 4 Experiment results on ChemProt and BC5CDR.**

| Model | ChemProt RE | BC5CDR NER |
|---|---|---|
| Sci-BERT | 84.61 ± 0.58 | <u>89.26 ± 0.22</u> |
| RoBERTa | 81.10 ± 0.95 | 86.93 ± 0.20 |
| BioBERT (+PubMed) | <u>85.48 ± 0.52</u> | <u>89.24 ± 0.35</u> |
| KV-PLM | 84.59 ± 0.59 | <u>89.00 ± 0.33</u> |
| KV-PLM* | <u>85.19 ± 0.55</u> | <u>89.17 ± 0.22</u> |

Underlined numbers denote the best scores with no significant difference.

the same as general text. In the other variant KV-PLM*, inspired by the SPE-based generative models[47], we apply BPE to SMILES strings to better control the tokenization of SMILES strings. For example, by controlling the vocabulary size of the SMILES string tokenizer, we can largely prevent over-specific infrequent tokenization results.

For the retrieval system, we regard SMILES strings as queries and descriptions retrieval candidates. The core idea is to focus on the nearest negative samples instead of all. The encoder for SMILES strings is the same one with descriptions since SMILES strings are also linear text and can be easily fused. Let $f(\mathbf{t})$ be a feature-based representation computed by encoder from text $\mathbf{t}$. Define the retrieval score between SMILES string of a molecule $\mathbf{m}$ and a unit of description $\mathbf{d}$ as:

$$s(\mathbf{m}, \mathbf{d}) = \frac{f(\mathbf{m}) \cdot f(\mathbf{d})}{|f(\mathbf{m})| \cdot |f(\mathbf{d})|}, \tag{1}$$

which is the cosine similarity of two representations. We refer to the loss function in VSE++[48] which is a similar representative image-caption retrieval method. For a positive pair $(\mathbf{m},\mathbf{d})$, we calculate the *Max of Hinges* (MH) loss:

$$\begin{aligned}\mathcal{L}_{\mathrm{MH}} = &\max_{\mathbf{d}'}[\alpha + s(\mathbf{m}, \mathbf{d}') - s(\mathbf{m}, \mathbf{d})]\\ &+ \max_{\mathbf{m}'}[\alpha + s(\mathbf{m}', \mathbf{d}) - s(\mathbf{m}, \mathbf{d})],\end{aligned} \tag{2}$$

where $\alpha$ is a margin hyperparameter, $\mathbf{d}'$ and $\mathbf{m}'$ are negative descriptions and SMILES strings from the batch.

**Baselines tailored for mono-information tasks**. For mono-information tasks including natural language tasks and molecule tasks, there are plenty of mature methods specially designed for them. Here, we compare our models with baselines tailored for mono-information tasks.

Experiment results on 4 MoleculeNet themes are shown in Table 3. D-MPNN[8] is a supervised graph convolution based method combined with descriptors. Random forest (RF)[49] is a representative method for statistical machine learning which also takes descriptors as the input. DMP[50] is an unsupervised pre-training method that takes SMILES strings and molecular graphs as the input.

Although not tailored for molecule tasks, our models still achieve reasonable performance compared to strong baselines. It is promising to leverage more advanced molecule encoders in PLMs to further improve the results, which we leave for future research.

Experiment results for ChemProt relation extraction and BC5CDR NER are shown in Table 4. We observe that pre-trained language models are generally the best solutions for these natural language processing tasks. We report the results of BioBERT (+PubMed) and RoBERTa[51], which are both popular models and achieve comparable results with Sci-BERT. Note that the initial version of BioBERT[36] underperforms Sci-BERT, while the recently released version is additionally trained on PubMed corpus, which helps it become the state-of-the-art model on ChemProt and BC5CDR. Our models achieve comparable performance with BioBERT (+PubMed).

**Training settings**. For Natural Language tasks, the authors of BERT provided range of possible values to work well across various tasks: batch size [8, 16, 32], Adam learning rate [$2e-5$, $3e-5$, $5e-5$], epoch number [2, 3, 4]. We conduct grid search in the hyper-parameters above. In ChemProt RE task, we set batch size as 8, learning rate as $2e-5$ and epoch number as 4. In BC5CDR NER task, we set batch size as 16, learning rate as $3e-5$ and epoch number as 4. Grid search is also done for the strongest baseline model Sci-BERT and the best hyper-parameters are proved to be the same.

For the MoleculeNet tasks, we search suitable learning rate in [$5e-6$, $5e-5$, $5e-4$] and batch size in [64, 128, 256], which are bigger than above because data points are more than natural sentences. It turns out to be relatively insensitive to hyperparameters changes as long as convergence is guaranteed. In MoleculeNet tasks, we set batch size as 128, learning rate as $5e-6$ and epoch number as 20. In USP-few task, we set batch size as 256, learning rate as $5e-5$ and epoch number as 30.

For retrieval training, since the batch size recommended in VSE++[48] is 128 while our training data scale is not so huge, we finally set batch size as 64. We set epoch number as 30, learning rate as $5e-5$ and margin as 0.2.

Notice that in all the experiments above, BertAdam optimizer is used and warmup proportion is 0.2. Max length is 128 for sentences and 64 for SMILES strings. For rxnfp model, since it is the only model of which the hidden size is 256 instead of 768, we set the learning rate as $5e-4$ due to the smaller scale of parameters and get better performance.

Tools and packages that we used in the experiments include: torch, transformers, numpy, sklearn, tqdm, seqeval, chainer-chemistry, rdkit, subword-nmt, boto3, and requests.

**Reporting summary**. Further information on research design is available in the Nature Research Reporting Summary linked to this article.

## Data availability

Data that support the findings of this study have been deposited in Google Drive: https://drive.google.com/drive/folders/1xig3-3JG63kR-Xqj1b9wkPEdxtfD_4IX.

## Code availability

The code of this study[52] can be obtained from GitHub: https://github.com/thunlp/KV-PLM. The zip file of code can be downloaded via the Google Drive link above.

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

## Acknowledgements

All authors are supported by the National Key Research and Development Program of China (No. 2020AAA0106501) and a grant from the Institute Guo Qiang, Tsinghua University (No. 2019GQB0004).

## Author contributions

Z.Z. and Y.Y. contributed to the conception of the study and wrote the manuscript; Z.Z. implemented the model framework and performed the experiment; Z.L. and M.S. led and provided valuable advice to the research.

## Competing interests

The authors declare no competing interests.
