## [Peer Review File · Nature Communications]

Reviewers' Comments:

Reviewer #1:

Remarks to the Author:

Summary: The authors present a Deep Learning system that learns both modalities, text and chemical structure, together.

General comments: The presented approach is novel and the main idea is scientifically appealing. The topic is relevant for the drug discovery field. However, there are severe technical errors and flaws. Several related scientific areas and works are ignored. The technical description is extremely lacking. The work is currently not reproducible.

Reproducibility: Neither the code nor the data is provided by the authors such that this work cannot be reproduced.

Major comments:

A) Novelty: The approach to co-learn representations of molecules and biomedical texts is novel. Nevertheless, the authors should embed their work into other works, where multi-modal Deep Learning systems are employed.

a) Medical images and text: reference [1].

b) Natural images and text: reference [2].

c) Protein sequences and molecules: reference [3].

d) Chemical reactions and molecules: reference [4].

e) Text and molecules: reference [5].

The authors should embed their work into related works with similar approaches and acknowledge prior work in appropriate form.

B) Relevance: This work represents a step towards human-level understanding of chemical structures and offers the opportunity to interact with this knowledge via natural language.

Therefore, this work is relevant in the area of drug discovery and biomedicine. The work is less relevant in the area of machine learning, since this represents a collection of existing techniques.

The authors should state their contributions and relevance clearer at the end of the Introduction section.

C) Technical errors and flaws:

a) The description of the prediction tasks are severely lacking. For almost all tasks it is unclear what the prediction task is, where the labels are from, what the size of the dataset is, how the training, validation, and test sets are put together, what the evaluation metric is, what the compared methods are and how hyperparameters are selected.

The authors should provide for each prediction task, at least the following:

Description of the dataset: number of data points (train, val, test), nature of the prediction task, evaluation metrics,

state-of-the-art methods, compared methods, hyperparameter selection strategy.

b) For the molecular property prediction tasks, the predictive quality is extremely low and far away from any state of the art. Since the method

has not been designed for property prediction, these experiments are also not relevant. The authors remove task on one modality, clearly state that their method is proficient at combining natural language and chemical structure and thus focus on the "versatile reading tasks", which is impressive enough.

c) Inappropriate and missing compared methods. The authors only compare their method against other methods that they implemented and thus ignore a huge set of methods for the specific tasks. For molecular property prediction, there are at least 10 methods can easily be compared, including Random Forests, Gradient Boosting, Support Vector Machines, multi-task neural networks, message-passing neural networks, etc (e.g. see [6]) Similarly, for chemical reaction classification, there are tens of methods, see (e.g. methods named in [7] or [4]). Again, these special tasks should rather be removed from the paper since the presented method is inferior to other methods in the field.

d) Missing error bars and statistical tests. All performance metrics are reported without error bars and comparisons without statistical test. In the current form, the performance values or purported improvement could just arise by chance. The authors should perform re-runs or cross-validation for each method to obtain reliable estimates and confidence intervals for their reported metrics.

e) The comparison is biased towards the authors' method in multiple ways. First, the authors use two variants of their own method against single variants of compared methods, which makes the comparison unfair. Second, it appears that the hyperparameters of the authors' method have been adapted to the specific task, whereas the hyperparameters of compared methods have not.

f) The architecture and all hyperparameters of the method are not described. The authors should clearly describe their architecture with all important parameters. Also the most important hyperparameters of the method should be stated and how these were adapted. In the current form, this central part is completely absent.

g) Ad-hoc decisions and unjustified choices. The authors propose an architecture that contains many different elements, such as the use of SMILES strings, segmenting with BPE, pre-training strategy, text model. While the text model (BERT) is justified by it being one of the best and most widely used models, the other choices are particularly poor, ad-hoc and unjustified. The best performing molecule encoders appear to be based on chemical descriptors plus a fully-connected network or message-passing networks in combination with chemical descriptors ([6,8,9]). BERT-based deep architectures are inferior to those [10]. Also the tokenization step of the SMILES string is an arbitrary and potentially constraining choice. The authors should justify their choices on the architecture either by citing work that backs their choice or by presenting data.

h) The description of the preprocessing steps for the datasets are lacking. For example, the description of the corpus S2orc and PubChem: which documents are used? How were they obtained from PubChem. What were the filtering criteria? The authors should strongly improve the description of the preprocessing of all datasets, such that this work can be reproduced. Furthermore,

the datasets should be provided as supplementary material.

i) The description of the comparison against human experts is completely missing (see major comment F).

D) Related areas and works ignored: see Major Comment A.

E) Technical description missing: see Major Comment C.

F) Lacking description on the comparison with human experts. It appears that the authors have performed some comparison of their system with human experts. However, this is a difficult task to ensure a fair comparison (if this can ever be done at all). What were the exact circumstances of this experiment? How many human experts? How recruited? How did they get questions? How was their answer required? Did they know that they were competing against an AI?

It is extremely concerning that these crucial details are missing in the manuscript.

Incorrect or tenuous claims:

a) "Molecule fingerprints computed by neural networks have achieved comparable or even better performance than expert-crafted descriptors": This is incorrect. Currently, neural networks built on expert-designed features or chemical fingerprints are on average better than pure graph neural networks or message-passing neural network at activity prediction tasks [8]. Citation [29] is especially misleading because this work combines the expert-crafted descriptors with message-passing neural nets which appear to give the method a slight edge over using each of those alone. The citations at this point are also strongly biased towards MIT/Stanford works. Therefore, the authors should re-write this paragraph on "Related Work".

Questions to the authors:

A) What is the problem that you are actually trying to solve with this approach? Can you state this clearer?

Minor comments:

a) The Tox21 dataset of MoleculeNet is an incorrect version of the original Tox21 dataset (data splits, number of molecules) and should be replaced by the original one.
None further at this stage.

Typos:

None at this stage.

References

- [1] Zhang, Y., Jiang, H., Miura, Y., Manning, C. D., & Langlotz, C. P. (2020). Contrastive learning of medical visual representations from paired images and text. arXiv preprint arXiv:2010.00747.
- [2] Radford, A., Kim, J. W., Hallacy, C., Ramesh, A., Goh, G., Agarwal, S., ... & Sutskever, I. (2021). Learning transferable visual models from natural language supervision. arXiv preprint arXiv:2103.00020.
- [3] Lenselink, E. B., Ten Dijke, N., Bongers, B., Papadatos, G., Van Vlijmen, H. W., Kowalczyk, W., ... & Van Westen, G. J. (2017). Beyond the hype: deep neural networks outperform established methods using a ChEMBL bioactivity benchmark set. *Journal of cheminformatics*, 9(1), 1-14.
- [4] Seidl, P., Renz, P., Dyubankova, N., Neves, P., Verhoeven, J., Wegner, J. K., ... & Klambauer, G. (2021). Modern Hopfield Networks for Few-and Zero-Shot Reaction Prediction. arXiv preprint arXiv:2104.03279.
- [5] Copara, J., Naderi, N., Knafou, J., Ruch, P., & Teodoro, D. (2020). Named entity recognition in chemical patents using ensemble of contextual language models. arXiv preprint arXiv:2007.12569.
- [6] Mayr, A., Klambauer, G., Unterthiner, T., Steijaert, M., Wegner, J. K., Ceulemans, H., ... & Hochreiter, S. (2018). Large-scale comparison of machine learning methods for drug target

prediction on ChEMBL. *Chemical science*, 9(24), 5441-5451.

[7] Coley, C. W., Green, W. H., & Jensen, K. F. (2018). Machine learning in computer-aided synthesis planning. *Accounts of chemical research*, 51(5), 1281-1289.

[8] Jiang, D., Wu, Z., Hsieh, C. Y., Chen, G., Liao, B., Wang, Z., ... & Hou, T. (2021). Could graph neural networks learn better molecular representation for drug discovery? A comparison study of descriptor-based and graph-based models. *Journal of cheminformatics*, 13(1), 1-23.

[9] Yang, K., Swanson, K., Jin, W., Coley, C., Eiden, P., Gao, H., ... & Barzilay, R. (2019). Analyzing learned molecular representations for property prediction. *Journal of chemical information and modeling*, 59(8), 3370-3388.

[10] Chithrananda, S., Grand, G., & Ramsundar, B. (2020). Chemberta: Large-scale self-supervised pretraining for molecular property prediction. *arXiv preprint arXiv:2010.09885*.

Reviewer #2:

None

Reviewer #3:

Remarks to the Author:

The authors have developed a method to acquire knowledge about molecules from mining biomedical papers.

While in principle novel and interesting, I think the paper needs to be much more thoroughly written to be acceptable for publication.

Some examples of issues.

A striking issue is that already in the title it is mentioned that the system outperforms humans. However, besides stating the human accuracy is 0.72, there is no other information the human experiment.

I'm surprised the the CHEBI ontology that covers substructures and could be used for the same task as in the paper is not even mentioned.

Another potential issue that isn't mentioned is SMILES standardization including canonicalization, tautomer assignment etc that can have significant impact on the results.

It is not clear what results in Table 2 and Figure 5 that couldn't be identified through NER without using SMILES strings.

The statement that the code will be provided soon is fairly meaningless.

Response Letter: A Deep-learning System Bridging Molecule Structure and Biomedical Text with Comprehension Comparable to Human Professionals

We make some notable changes to the manuscript, supplementing important contents including: (1) Related works. We include more related works to better embed our work in the literature, clarify our main contributions, and support our model design choices. (2) Detailed descriptions. We add detailed explanations about the datasets, models, training process, and human evaluation. (3) Error bars and statistical tests. We rerun the experiments and report the error bars and results of statistical tests. The main conclusion of our work has not been changed.

In the following, the comments of the reviewers are shown in blue and our responses are shown in black. The revised contents have also been highlighted in blue in the manuscript to help track the changes.

Response to the comments of Reviewer 1

1. **Reproducibility:** Neither the code nor the data is provided by the authors such that this work cannot be reproduced.

Thanks for the comment. We provide data and code download links in the data and code availability section in the revised manuscript.

2. **Novelty:** The approach to co-learn representations of molecules and biomedical texts is novel. Nevertheless, the authors should embed their work into other works, where multi-modal Deep Learning systems are employed. The authors should embed their work into related works with similar approaches and acknowledge prior work in appropriate form.

We add citations and discussions about multi-

modal deep learning systems according to the suggestion in the related works section.

3. **Relevance:** This work represents a step towards human-level understanding of chemical structures and offers the opportunity to interact with this knowledge via natural language. Therefore, this work is relevant in the area of drug discovery and biomedicine. The work is less relevant in the area of machine learning, since this represents a collection of existing techniques. The authors should state their contributions and relevance clearer at the end of the Introduction section.

We agree that our work is more relevant in the area of drug discovery and biomedicine instead of machine learning. Contributions are re-summarized and added at the end of the introduction section.

4. The description of the prediction tasks are severely lacking. For almost all tasks it is unclear what the prediction task is, where the labels are from, what the size of the dataset is, how the training, validation, and test sets are put together, what the evaluation metric is, what the compared methods are and how hyperparameters are selected. The authors should provide for each prediction task, at least the following: Description of the dataset: number of data points (train, val, test), nature of the prediction task, evaluation metrics, state-of-the-art methods, compared methods, hyperparameter selection strategy.

We add more details about the tasks and datasets, including the description of tasks and labels, number

of data points, splitting method, download link, nature of the prediction tasks, and evaluation metrics. Hyperparameter selection is explained in the training setting section. We also include more baselines for mono-domain tasks.

5. For the molecular property prediction tasks, the predictive quality is extremely low and far away from any state of the art. Since the method has not been designed for property prediction, these experiments are also not relevant. The authors remove task on one modality, clearly state that their method is proficient at combining natural language and chemical structure and thus focus on the "versatile reading tasks", which is impressive enough.

Thanks for the comments that versatile reading tasks are impressive enough. We present molecular property prediction tasks mainly for the comparison between our models and the baseline models such as Sci-BERT to prove that, although still far from state-of-the-art, our knowledge inserting pre-training method can also improve PLMs capability in molecule understanding. More tailored baselines and the state-of-the-art method are further compared in the method section. Although KV-PLM is not designed for these tasks, it outperforms some classical baseline models, which may inspire future works. Therefore, we would like to remain this part after careful consideration.

6. Inappropriate and missing compared methods. The authors only compare their method against other methods that they implemented and thus ignore a huge set of methods for the specific tasks. For molecular property prediction, there are at least 10 methods can easily be compared, including Random Forests, Gradient Boosting, Support Vector Machines, multi-task neural networks, message-passing neural networks, etc (e.g. see [6]) Similarly, for chemical reaction classification, there are tens of methods, see (e.g. methods named in [7] or [4]). Again, these special tasks should rather be removed from the paper since the presented method is inferior to other methods in the field.

Please refer to question 5.

7. Missing error bars and statistical tests. All performance metrics are reported without error bars and comparisons without statistical test. In the current form, the performance values or purported improvement could just arise by chance. The authors should perform re-runs or cross-validation for each method to obtain reliable estimates and confidence intervals for their reported metrics.

Thanks for the suggestion. We actually reported average results for several random seeds in the initial script. In the revision, we further unify the experimental settings and provide complete error bars for all the results to better support the conclusions.

8. The comparison is biased towards the authors' method in multiple ways. First, the authors use two variants of their own method against single variants of compared methods, which makes the comparison unfair. Second, it appears that the hyperparameters of the authors' method have been adapted to the specific task, whereas the hyperparameters of compared methods have not.

Thanks for the comment. (1) Two variants. We report the results of two variants since they are both plausible in design choices, and exhibit different strengths and characteristics, while both variants are overall stronger than the baselines. (2) Hyperparameters. In our initial manuscript, we adopt the same hyperparameters for all models when the model sizes and tasks are the same, since in this case, hyperparameters typically transfer well across models. In the revision, we further grid search hyperparameters for our strongest baseline (Sci-BERT) and update the results in Table 1. The changes in baseline performance after grid search are marginal as expected, and the conclusions still hold.

9. The architecture and all hyperparameters of the method are not described. The authors should clearly describe their architecture with all important parameters. Also the most important hyperparameters of the method should be stated and how these were adapted. In the current form, this central part is completely absent.

Thanks for pointing out the issue. We have added the details about the architecture and adaptation methods in the model subsection, and described the best hyperparameters and the selection approach in the training settings subsection.

10. Ad-hoc decisions and unjustified choices. The authors propose an architecture that contains many different elements, such as the use of SMILES strings, segmenting with BPE, pre-training strategy, text model. While the text model (BERT) is justified by it being one of the best and most widely used models, the other choices are particularly poor, ad-hoc and unjustified. The best performing molecule encoders appear to be based on chemical descriptors plus a fully-connected network or message-passing networks in combination with chemical descriptors ([6,8,9]). BERT-based deep architectures are inferior to those [10]. Also the tokenization step of the SMILES string is an arbitrary and potentially constraining choice. The authors should justify their choices on the architecture either by citing work that backs their choice or by presenting data.

Thanks for the comment. (1) Molecule encoder. Indeed, some simple structures designed for molecule encoding can solve this problem quite well. These years, deep architectures such as BERT are also widely used for it [1, 2, 3], greatly advancing the performance of the work mentioned in this comment [4]. Therefore, we believe the BERT-based molecule encoder is promising in encoding molecule structures. (2) SMILES tokenization. There are several previous works trying to tokenize the SMILES strings with BPE [5], and we cite some related work in the revision to back our choice. We also compare with alternatives to further support the design. SmilesPE [6] is a good tokenizer for SMILES string tasks based on BPE. We have experimented with SmilesPE and find that the model performs not so well on mono-information tasks (e.g., 88.95 for NER and 47.41 for USPTO-few, which are significantly lower than current results). One possible reason is that the tokenized molecule representations are over-specific, which may cause too many rarely used tokens inserted in general text and break the normal

structure of sentences. Therefore, we simply use BPE and set the vocabulary size relatively small, ensuring that tokenized molecule representations are commonly used and it becomes easier to train KV-PLM.

11. The description of the preprocessing steps for the datasets are lacking. For example, the description of the corpus S2orc and PubChem: which documents are used? How were they obtained from PubChem. What were the filtering criteria? The authors should strongly improve the description of the preprocessing of all datasets, such that this work can be reproduced. Furthermore, the datasets should be provided as supplementary material.

Thanks for the suggestion. We have added detailed descriptions of the corpus and download links of datasets in the corpus and dataset section. PCdes is crawled from PubChem with the restriction that the chemical needs to have at least a paragraph of text descriptions. Preprocessing information is also provided in the revision.

12. Lacking description on the comparison with human experts. It appears that the authors have performed some comparison of their system with human experts. However, this is a difficult task to ensure a fair comparison (if this can ever be done at all). What were the exact circumstances of this experiment? How many human experts? How recruited? How did they get questions? How was their answer required? Did they know that they were competing against an AI? It is extremely concerning that these crucial details are missing in the manuscript.

It is a crucial problem in our initial script. We have added a description of human expert experiments in the versatile reading subsection. We recruited more experts in human evaluation for more reliable results. It is indeed a difficult task to ensure fairness, and we tried to maximally reach this goal.

13. Incorrect or tenuous claims: "Molecule fingerprints computed by neural networks have achieved comparable or even better performance than expert-crafted descriptors": This is incorrect. Currently,

neural networks built on expert-designed features or chemical fingerprints are on average better than pure graph neural networks or message-passing neural network at activity prediction tasks [8]. Citation [29] is especially misleading because this work combines the expert-crafted descriptors with message-passing neural nets which appear to give the method a slight edge over using each of those alone. The citations at this point are also strongly biased towards MIT/Stanford works. Therefore, the authors should re-write this paragraph on "Related Work".

We are very grateful to the reviewer for pointing out this mistake. The related claim has already been corrected in the new script.

14. Questions to the authors: What is the problem that you are actually trying to solve with this approach? Can you state this clearer?

We are trying to solve the isolation problem of the current machine learning models when processing materials from biomedical area. Moreover, by bridging molecule structures and biomedical text, our model can be applied to drug discovery and documentation to assist biomedical research.

15. Minor comments: The Tox21 dataset of MoleculeNet is an incorrect version of the original Tox21 dataset (data splits, number of molecules) and should be replaced by the original one.

Since Tox21 is used as a sub-task in the MoleculeNet benchmark, we download and use the latest version from the official homepage of MoleculeNet instead of the original one for the convenience of comparison.

Response to the comments of Reviewer 2

1. There should be multiple ways to describe a molecule structure, like describing the structure, the function, its solubility, its toxicity, etc. My concern here is which way the model chooses to describe the structure? How does the model control the generated text with the given molecule structures?

This is an insightful comment. The molecule description is actually retrieved from the description

base instead of generated at this stage. The output text of our model will follow the distribution of the training data used (i.e., molecule descriptions written by human experts in knowledge base). For example, our model may tend to choose to describe the toxicity of molecules if the descriptions written by human experts in the knowledge base pay more attention to toxicity. We believe this property of our model is reasonable and helpful in real-world applications, since in this way our model can produce descriptions that human experts are most interested in. More details are explained in Versatile Reading and Case Study in the Results section.

2. This type of pre-training strategy with cross-information is not new. Previous studies already jointly exploit text and image to train deep learning models, such as multi-modal pretraining models in "LayoutLMv2: Multi-modal pre-training for visually-rich document understanding" and "M3p: Learning universal representations via multitask multilingual multimodal pre-training".

Thanks for the comment. Although bridging and grounding molecule structure and general text in PLMs is a new idea, we have indeed been inspired by plenty of visual-language cross-modal researches, and even refer to the classic VSE++ method in visual-language embeddings for our retrieval system. We agree it will be beneficial to discuss more related works on cross-modal pre-training in the paper to better embed our work in the literature. In our revision, we have added more citations related to cross-modal pre-training in Related Work in the Method section.

3. More notable baseline models are supposed to be compared with, such as BioBERT, RoBERTa, etc.

Thanks for the suggestion. The original version of BioBERT is beaten by Sci-BERT according to the results provided by the original papers and therefore we choose Sci-BERT as our baseline. We have added results for the latest version of BioBERT (+PubMed), which is further pre-trained on PubMed corpus and is the state-of-the-art model for ChemProt and BC5CDR in Table 4 in the

Method section. The result for RoBERTa is also reported as a representative of better general domain PLMs.

Response to the comments of Reviewer 3

1. A striking issue is that already in the title it is mentioned that the system outperforms humans. However, besides stating the human accuracy is 0.72, there is no other information the human experiment.

Thanks for the comments. It is a crucial problem in our initial manuscript. We have added the description of human expert experiments in Versatile Reading subsection. We recruited more experts in human evaluation for more reliable results.

2. I'm surprised the the CHEBI ontology that covers substructures and could be used for the same task as in the paper is not even mentioned.

CHEBI is not mentioned separately because PubChem crawls useful information from many websites including CHEBI. (e.g. Salicylic acid¹)

3. Another potential issue that isn't mentioned is SMILES standardization including canonicalization, tautomer assignment etc that can have significant impact on the results.

Thanks for reminding. During pre-training, the SMILES strings from PubChem are all canonicalized, while there is no special treatment for other strings from downstream tasks for generalization and simplicity.

4. It is not clear what results in Table 2 and Figure 5 that couldn't be identified through NER without using SMILES strings.

We are really sorry that we have not totally understood this comment. Using SMILES strings should not cause information leakage in NER, since most of the entities in ChemProt dataset are expressed in their textual names instead of SMILES strings. SMILES strings are only inserted in a part of pre-training corpus to strengthen the model capability of understanding chemicals, while we just

¹<https://pubchem.ncbi.nlm.nih.gov/compound/338>

use the original normal sentences in downstream natural language tasks.

5. The statement that the code will be provided soon is fairly meaningless.

Thanks for pointing out the issue. We provide code and data download links at the end of the paper this time.

References

- [1] S. Wang, Y. Guo, Y. Wang, H. Sun, and J. Huang, "Smiles-bert: large scale unsupervised pre-training for molecular property prediction," in *Proceedings of the 10th ACM international conference on bioinformatics, computational biology and health informatics*, pp. 429–436, 2019.
- [2] S. Honda, S. Shi, and H. R. Ueda, "Smiles transformer: pre-trained molecular fingerprint for low data drug discovery," *arXiv preprint arXiv:1911.04738*, 2019.
- [3] S. Lim and Y. O. Lee, "Predicting chemical properties using self-attention multi-task learning based on smiles representation," *arXiv preprint arXiv:2010.11272*, 2020.
- [4] S. Chithrananda, G. Grand, and B. Ramsundar, "Chemberta: Large-scale self-supervised pre-training for molecular property prediction," *arXiv preprint arXiv:2010.09885*, 2020.
- [5] Y. Omote, K. Matsushita, T. Iwakura, A. Tamura, and T. Ninomiya, "Transformer-based approach for predicting chemical compound structures," in *Proceedings of the 1st Conference of the Asia-Pacific Chapter of the Association for Computational Linguistics and the 10th International Joint Conference on Natural Language Processing*, pp. 154–162, 2020.
- [6] X. Li and D. Fourches, "Smiles pair encoding: A data-driven substructure tokenization algorithm for deep learning," *Journal of Chemical Information and Modeling*, vol. 61, no. 4, pp. 1560–1569, 2021.

Reviewers' Comments:

Reviewer #1:

Remarks to the Author:

Summary: The authors present a Deep Learning system that learns both modalities, text and chemical structure, together.

This is the review of the first revision of the paper.

General comments: The presented approach is novel and the main idea is scientifically appealing. The topic is relevant for the drug discovery field.

With the new version several technical errors have been corrected and the manuscript has strongly improved.

However, there are two main concerns that remain (major comments), such that I recommend a major revision.

Reproducibility: Reproducibility has been strongly improved due to the availability of the code and data.

Positive aspects:

The work approaches a relevant problem in biology, medicine and molecular biology and related fields. The presented computational approach is reasonably novel and leads to interesting results and potentially open new ways for practitioners and researchers to acquire knowledge.

Major comments:

A) The choice of the molecular encoder is still an ad-hoc decision and a strongly limiting factor of this work. The authors chose to tokenize the SMILES strings and then use this as input for their BERT model. However, any other molecule encoder would be a suitable choice. A belief that a BERT-based molecule encoder is good is a poor argument against the amount of literature that stands against it (at least [1,2,3,4]-- in none of those a Smiles-transformer based method is the best method). At the least, the authors should state this choice as a strongly limiting factor and a decision made for programmatic simplicity at a prominent position in the manuscript.

B) Experiments for molecular property/activity prediction and Table 3 are still incorrect and strongly misleading.

a) As stated in my previous review, for Tox21 there is a pre-defined test set (challenge test set), on which methods are compared. The authors pick a 8:1:1 random split, which means they defined their own test set, such that performance values cannot be compared. The authors should use the pre-defined test set of Tox21 to assess the performance of their method. The error also becomes immediately evident by their reported performance of an AUC of 0.90, while the SOTA is at 0.871 [5].

b) There must be a similar problem with the calculation of the performance values for the datasets BBBP, HIV, and SIDER,

since the authors report values far above the state of the art and all reference values. Concretely, for BBBP the best methods usually perform around 0.92-0.93, for HIV around 0.83-0.84, for SIDER around 0.67-0.68 (see eg. [2],[6],[7],[8]), whereas the authors report 0.988, 0.825, and 0.858. Especially for SIDER, such an AUC is hardly possible based on the characteristics of the dataset. The authors should re-visit their results on these datasets and check for correct model and hyperparameter selection or possible data leakage.

c) The authors introduce a "false balancedness" by their choice of baselines ("GraphConv", "TextCNN", "SA-MTL"), i.e., it appears as if those were the standard methods for this type of datasets. However, the standard would be descriptor-based multi-task deepnetworks (e.g.[3]), descriptor-based Random Forests, or graph-convolutions combined with descriptors [1]. The authors should report performance values obtained with standard baselines, and the current state-of-the-art at these datasets and show comparable performance values of their own compared methods.

d) The reviewer appreciates the point that the authors want to make about their model that it is also capable of being used for molecular property prediction, but at this point the experiments are so flawed (points a,b,c), such these experiments should rather be removed from the manuscript.

Minor comments:

Formulas need mathematical typesetting.

References:

- [1] Yang, K., Swanson, K., Jin, W., Coley, C., Eiden, P., Gao, H., ... & Barzilay, R. (2019). Analyzing learned molecular representations for property prediction. *Journal of chemical information and modeling*, 59(8), 3370-3388.
- [2] Jiang, D., Wu, Z., Hsieh, C. Y., Chen, G., Liao, B., Wang, Z., ... & Hou, T. (2021). Could graph neural networks learn better molecular representation for drug discovery? A comparison study of descriptor-based and graph-based models. *Journal of cheminformatics*, 13(1), 1-23.
- [3] Mayr, A., Klambauer, G., Unterthiner, T., Steijaert, M., Wegner, J. K., Ceulemans, H., ... & Hochreiter, S. (2018). Large-scale comparison of machine learning methods for drug target prediction on ChEMBL. *Chemical science*, 9(24), 5441-5451.
- [4] Wu, Z., Ramsundar, B., Feinberg, E. N., Gomes, J., Geniesse, C., Pappu, A. S., ... & Pande, V. (2018). MoleculeNet: a benchmark for molecular machine learning. *Chemical science*, 9(2), 513-530.
- [5] Alperstein, Z., Cherkasov, A., & Rolfe, J. T. (2019). All smiles variational autoencoder. *arXiv preprint arXiv:1905.13343*.
- [6] Ramsauer, H., Schäfl, B., Lehner, J., Seidl, P., Widrich, M., Adler, T., ... & Hochreiter, S. (2020). Hopfield networks is all you need. *arXiv preprint arXiv:2008.02217*.
- [7] <https://paperswithcode.com/sota/drug-discovery-on-sider>
- [8] Stepišnik, T., Škrlj, B., Wicker, J., & Kocev, D. (2021). A comprehensive comparison of molecular feature representations for use in predictive modeling. *Computers in Biology and Medicine*, 130, 104197.

Reviewer #3:

Remarks to the Author:

My comments have been satisfactorily addressed so I would support publication

Response Letter: A Deep-learning System Bridging Molecule Structure and Biomedical Text with Comprehension Comparable to Human Professionals

Thanks for the careful review of all the editors and reviewers. We make the following revision to the manuscript: (1) MoleculeNet experiments. We get the split data from the DeepChem package and revise the related results. Baseline methods are also changed. (2) Explanation for the SMILES encoder. We supplement the reason for choosing SMILES as the molecule encoding method in the introduction.

In the following, the comments of the reviewer 1 are shown in blue and our responses are shown in black. The revised contents have also been highlighted in blue in the manuscript to help track the changes.

Response to the comments of Reviewer 1

A) The choice of the molecular encoder is still an ad-hoc decision and a strongly limiting factor of this work. The authors chose to tokenize the SMILES strings and then use this as input for their BERT model. However, any other molecule encoder would be a suitable choice. A belief that a BERT-based molecule encoder is good is a poor argument against the amount of literature that stands against it (at least [1,2,3,4]– in none of those a Smiles-transformer based method is the best method). At the least, the authors should state this choice as a strongly limiting factor and a decision made for programmatic simplicity at a prominent position in the manuscript.

Thanks for the comment. We carefully read the references you provided and accepted your adjustment suggestions, supplementing the programmatic simplicity statement in the introduction.

B) Experiments for molecular property/activity prediction and Table 3 are still incorrect and strongly misleading.

a) As stated in my previous review, for Tox21 there is a pre-defined test set (challenge test set), on which methods are compared. The authors pick a 8:1:1 random split, which means they defined their own test set, such that performance values cannot be compared. The authors should use the pre-defined test set of Tox21 to assess the performance of their method. The error also becomes immediately evident by their reported performance of an AUC of 0.90, while the SOTA is at 0.871.

b) There must be a similar problem with the calculation of the performance values for the datasets BBBP, HIV, and SIDER, since the authors report values far above the state of the art and all reference values. Concretely, for BBBP the best methods usually perform around 0.92-0.93, for HIV around 0.83-0.84, for SIDER around 0.67-0.68, whereas the authors report 0.988, 0.825, and 0.858. Especially for SIDER, such an AUC is hardly possible based on the characteristics of the dataset. The authors should re-visit their results on these datasets and check for correct model and hyperparameter selection or possible data leakage.

Thanks for the suggestion. We check the data again and there is no data leakage problem. For the high score of SIDER and Tox21, we check the source codes of some baselines and find that there exist different evaluation methods for the two multi-label datasets. In the last manuscript, we treat mul-

multiple tasks as a whole to calculate the AUC score. In the current version, following [1], the results are replaced by the more common average AUC scores of the multiple tasks. The adjusted results meet the reasonable range mentioned by the reviewer.

For the split of the dataset, we find there are several common practices in the literature: The method that MoleculeNet [2] requires is to divide the data into 8:1:1 by scaffold split (for BBBP and HIV) or random split (for SIDER and Tox21). Many works [3, 4], including D-MPNN [5] that the reviewer mentioned, also follow this method and define their test set. Some other works take 5-fold cross-validation instead [6, 7]. Since the reviewer suggests us to use the pre-defined test set and refer to the Tox21 SOTA method [1], we follow its setting and adopt the official train, valid and test sets in the DeepChem [8] package to ensure fairness, and revise our results in the latest version.

c) The authors introduce a "false balancedness" by their choice of baselines ("GraphConv", "TextCNN", "SA-MTL"), i.e., it appears as if those were the standard methods for this type of datasets. However, the standard would be descriptor-based multi-task deepnetworks, descriptor-based Random Forests, or graph-convolutions combined with descriptors. The authors should report performance values obtained with standard baselines, and the current state-of-the-art at these datasets and show comparable performance values of their own compared methods.

d) The reviewer appreciates the point that the authors want to make about their model that it is also capable of being used for molecular property prediction, but at this point the experiments are so flawed (points a,b,c), such these experiments should rather be removed from the manuscript.

Thanks for the suggestion about the choice of baselines. We adopt the graph-convolutions and Random Forest combined with descriptors as baseline models [5]. As the reviewer suggests, they are representatives of mainstream methods. Since the original paper [5] does not adopt the pre-defined test set in DeepChem, we turn to the results on the pre-defined DeepChem split reported in DMP [9]. It is a

relatively strong baseline under DeepChem setting. Meanwhile, DMP is an unsupervised method that takes the SMILES strings and molecular graphs as the input and the pre-training process effectively improves the performance. We take DMP as another baseline model to enrich the comparison. The DMP result can also be deleted if there are any misleading points. The above methods are all open source so that reproducibility is guaranteed.

We sincerely appreciate the suggestion about removing the molecule experiments, and the performance of our model is indeed not good enough especially after the evaluation settings are revised. Nevertheless, after careful consideration and discussion, we would like to improve the experiments by including more representative baselines and adopting more widely used evaluation settings, instead of removing this part of the content. Otherwise, the claim of versatile capability may be less persuasive. Besides, we believe that reporting the results that need to be further improved can also inspire related discussion and future research. We hope this decision will gain your approval.

References

- [1] Z. Alperstein, A. Cherkasov, and J. T. Rolfe, "All smiles variational autoencoder," *arXiv preprint arXiv:1905.13343*, 2019.
- [2] Z. Wu, B. Ramsundar, E. N. Feinberg, J. Gomes, C. Geniesse, A. S. Pappu, K. Leswing, and V. Pande, "Moleculenet: a benchmark for molecular machine learning," *Chemical science*, vol. 9, no. 2, pp. 513–530, 2018.
- [3] Z. Quan, X. Lin, Z.-J. Wang, Y. Liu, F. Wang, and K. Li, "A system for learning atoms based on long short-term memory recurrent neural networks," in *2018 IEEE International Conference on Bioinformatics and Biomedicine (BIBM)*, pp. 728–733, IEEE, 2018.
- [4] K. Abbasi, A. Poso, J. Ghasemi, M. Amanlou, and A. Masoudi-Nejad, "Deep transferable com-

- pound representation across domains and tasks for low data drug discovery,” *Journal of chemical information and modeling*, vol. 59, no. 11, pp. 4528–4539, 2019.
- [5] K. Yang, K. Swanson, W. Jin, C. Coley, P. Eiden, H. Gao, A. Guzman-Perez, T. Hopper, B. Kelley, M. Mathea, *et al.*, “Analyzing learned molecular representations for property prediction,” *Journal of chemical information and modeling*, vol. 59, no. 8, pp. 3370–3388, 2019.
- [6] X. Lin, Z. Quan, Z.-J. Wang, H. Huang, and X. Zeng, “A novel molecular representation with bigru neural networks for learning atom,” *Briefings in bioinformatics*, vol. 21, no. 6, pp. 2099–2111, 2020.
- [7] Y. Song, S. Zheng, Z. Niu, Z.-H. Fu, Y. Lu, and Y. Yang, “Communicative representation learning on attributed molecular graphs,” in *IJCAI*, pp. 2831–2838, 2020.
- [8] B. Ramsundar, *Molecular machine learning with DeepChem*. PhD thesis, Stanford University, 2018.
- [9] J. Zhu, Y. Xia, T. Qin, W. Zhou, H. Li, and T.-Y. Liu, “Dual-view molecule pre-training,” *arXiv preprint arXiv:2106.10234*, 2021.

Reviewers' Comments:

Reviewer #1:

Remarks to the Author:

My remaining comments and concerns about the molecular property prediction tasks have been addressed.

Point-by-point response to the reviewers

Reviewer #1 (Remarks to the Author):

My remaining comments and concerns about the molecular property prediction tasks have been addressed.

Response:

Thanks for your careful review and inspiring suggestions.